# Elucidating the Ability of CGRP to Modulate Microvascular Events in Mouse Skin

**DOI:** 10.3390/ijms232012246

**Published:** 2022-10-13

**Authors:** Ali A. Zarban, Hiba Chaudhry, João de Sousa Valente, Fulye Argunhan, Hala Ghanim, Susan D. Brain

**Affiliations:** 1Section of Vascular Biology and Inflammation, School of Cardiovascular and Metabolic Medicine & Sciences, BHF Centre of Research Excellence, Franklin-Wilkins Building, Waterloo Campus, King’s College London, London SE1 9NH, UK; 2Department of Pharmacology and Toxicology, College of Pharmacy, Jazan University, Jazan 45142, Saudi Arabia

**Keywords:** CGRP, mouse skin, sensory neuropeptides, oedema formation, neutrophil accumulation, RTX, αCGRPKO mice, BIBN4096BS, Carrageenan, TNFα

## Abstract

Oedema formation and polymorphonuclear leukocyte (neutrophil) accumulation are involved in both acute and chronic inflammation. Calcitonin gene-related peptide (CGRP) is a sensory neuropeptide that is released from stimulated sensory nerves. CGRP is a potent vasodilator neuropeptide, especially when administered to the cutaneous microvasculature, with a long duration of action. Here, we have investigated the ability of vasodilator amounts of CGRP to modulate oedema formation and neutrophil accumulation induced in the cutaneous microvasculature of the mouse. To learn more about the mechanism of action of endogenous CGRP, we have investigated the response to the inflammatory stimulants tumour necrosis factor alpha (TNFα) and carrageenan in three different murine models: a model where sensory nerves were depleted by resiniferatoxin (RTX); a pharmacological method to investigate the effect of a selective CGRP receptor antagonist; and a genetic approach using wildtype (WT) and αCGRP knockout (KO) mice. Our results show that exogenous CGRP potentiates oedema formation induced by substance P (SP) and TNFα. This is further supported by our findings from sensory nerve-depleted mice (in the absence of all neuropeptides), which indicated that sensory nerves are involved in mediating the oedema formation and neutrophil accumulation induced by TNFα, and also carrageenan in cutaneous microvasculature. Furthermore, endogenous CGRP was shown to contribute to this inflammatory response as carrageenan-induced oedema formation is attenuated in WT mice treated with the CGRP receptor antagonist, and in αCGRPKO mice. It is therefore concluded that CGRP can contribute to inflammation by promoting oedema formation in skin, but this response is dependent on the pro-inflammatory stimulus and circumstance.

## 1. Introduction

The 37 amino acid neuropeptide calcitonin gene-related peptide, CGRP, exists in two highly structurally related forms (α and β), both of which are potent vasodilators, with αCGRP considered to be the major form released from sensory nerves, often localised to the peripheral vasculature [1]. CGRP shares this potent microvascular activity with prostaglandins and nitric oxide [2]. However, evidence suggests that CGRP acts largely independently of these agents in vivo, even though CGRP has the ability to mediate vasodilation via a nitric oxide dependent relaxant mechanism on certain vessels in vitro [2,3,4]. CGRP has been predicted to have either pro- or anti-inflammatory activities, depending on the situation [5]. We and other groups have shown anti-inflammatory activities of CGRP, when CGRP is given systemically [6,7]. By comparison, there is also evidence from groups, including our own, [8,9] that CGRP can act in a pro-inflammatory manner in the cutaneous microvasculature in vivo and this is the subject of the present study. 

CGRP is co-localised with substance P (SP) in perivascular sensory nerves that are ideally situated to play a pivotal role in influencing oedema formation [5], a concept known as neurogenic inflammation [10]. SP acts via neurokinin1 receptors and also potentially via histamine released from mast cells, to increase microvascular permeability leading to oedema formation [8,11,12]. The increased permeability occurs at the post-capillary venule level and a range of inflammatory mediators such as bradykinin and histamine in addition to SP are capable of mediating this directly [13]. CGRP primarily acts to increase blood flow via the calcitonin receptor-like receptor (CLR) when complexed with a receptor activity modifying protein 1 (RAMP1), which is known as the canonical CGRP receptor [5]. CGRP, in vasodilator quantities, is not associated with oedema formation, when injected alone [8,9]. However, CGRP does have an ability to potentiate oedema formation induced by mediators of increased microvascular permeability [8,11,12]. The combined activities of CGRP and SP are shown in the present study and used as controls, where we examine the influence of CGRP on oedema formation and neutrophil accumulation induced by tumour necrosis factor alpha (TNFα) and carrageenan.

TNFα is perhaps one of the best-known inflammatory mediators as TNFα blocking agents show efficacy in a range of inflammatory diseases, including but not limited to, rheumatoid arthritis and psoriasis [14,15,16]. TNFα is a potent neutrophil attractant [17,18]. It is established that TNFα stimulates the upregulation of endothelial adhesion molecules such as vascular cell adhesion molecule-1 (VCAM-1) and intercellular adhesion molecule-1 (ICAM-1) that are instrumental in the time-dependent resulting neutrophil accumulation [19]. Neutrophils are also capable of mediating a neutrophil-dependent oedema phenomenon [20]. The seaweed extract carrageenan is also investigated, as it is an established inflammatory stimulus that can mediate acute and chronic inflammation following endogenous mediator generation [21]. In this study, we have investigated how exogenous and endogenous CGRP influences the oedema formation and neutrophil accumulation induced by TNFα and carrageenan in a murine model of inflammation [22,23]. 

We have used a multiple site murine model of cutaneous vascular inflammation where controls and test agents are injected intradermally (i.d.) into the same mouse, to reduce variation between mice [24]. Here we present evidence using three experimental strategies (depletion of sensory nerves, administration of a selective CGRP receptor antagonist (BIBN4096BS) and use of αCGRP knockout, KO, mice) to investigate the ability of CGRP to influence oedema formation and neutrophil accumulation.

## 2. Results

### 2.1. CGRP Potentiation of SP-Induced Oedema Formation over 30 min

The first experiment determined the acute oedema formation induced over 30 min by CGRP and SP, when injected intradermally (i.d.) (Figure 1a). Oedema formation was calculated by the local extravascular accumulation of Evans Blue (EB) in skin, denoted by the blue colouration. This is shown by volume, calculated using an equation based on the ellipsoid shape, observed at skin sites, following measurements of two diameters (mm × mm) and the depth of the blue colouration (mm) (Figure 1b). The major neutrophil-derived enzyme myeloperoxidase (MPO) was used as a biomarker for neutrophils [25]; the MPO activity (U/mL) is shown in Figure 1c. Control treatments were chosen to ensure that physical injury to the skin was not responsible for the observed responses. Minimal oedema formation or neutrophil accumulation was observed at the following sites: naïve skin site (un-injected), sham site (i.d. injection but no agent), vehicle site (Tyrode’s solution 50 µL i.d.). The test agents were given as follows: CGRP at an established vasodilator dose (20 pmol/50 µL i.d; [2,9]), which did not induce oedema formation, SP alone (300 pmol/50 µL i.d.), which induced oedema formation, and CGRP injected with SP as a co-administration (SP, 300 pmol/50 µL + CGRP, 20 pmol/50 µL i.d), which induced a synergistic response in terms of significant enhancement of SP-induced oedema formation, as previously shown (Figure 1c; [11]). Of note, the combined effect of SP + CGRP was higher than the sum of the CGRP and SP effects. The MPO activity (U/mL) for this experiment is shown in Figure 1c where the positive control (zymosan-induced peritoneal exudate) induced significant neutrophil accumulation and none of the sites showed an increase in MPO activity, in keeping with the concept that neither SP nor CGRP mediate acute neutrophil accumulation in mouse skin. This acute response was used as a positive oedema response in later experiments.

### 2.2. CGRP Potentiation of TNFα-Induced Oedema Formation over 4 h

TNFα stimulates a time-dependent accumulation of neutrophils, following the upregulation of adhesion molecules [19]. In order to investigate the effect of CGRP in TNFα-induced oedema and neutrophil accumulation over 4 h, we firstly carried out an experiment in which CGRP was injected at the last 30 min of the 4 h protocol (Figure 2a). As expected, CGRP had no effect when injected alone, and SP + CGRP induced oedema formation but not neutrophil accumulation (Figure 2b–d). The oedema response to TNFα was significantly potentiated when CGRP was injected i.d. at 3.5 h at the TNFα sites, whereas MPO activity was not significantly affected (Figure 2b–d). The combined effect of TNFα + CGRP was higher than the sum of the CGRP and TNFα effects. Following this result, we progressed to determine whether CGRP could potentiate neutrophil accumulation when added with TNFα at the beginning of the 4 h experimental period, but there was no change (Figure 2e,f).

### 2.3. The Effect of Depleting Sensory Nerves on TNFα and Carrageenan-Induced Oedema Formation and Neutrophil Accumulation

To deplete sensory nerves of their mediators that include SP and CGRP, we pre-treated mice with Resiniferatoxin (RTX) or vehicle solution (10% Tween-80, 10% ethanol, 80% saline s.c.) over four days (adapted from our studies on a murine model of psoriasis [26]). This technique is associated with side effects, so mice were pre-treated with a cocktail of anti-inflammatory agents (see methods) and the response to the RTX treatments were assessed by a severity scoring system (see Appendix A). The adverse effects (lack of mobility, responsiveness and grooming) were highest 1 h after RTX, with little effect noted in vehicle-pre-treated mice. Further monitoring of mice revealed that, by 4 h after RTX injection, they had recovered much of their well-being in terms of mobility and general appearance. The severity scoring system was also used on days 2–4 and a substantially lower scoring was noted. By day four of the RTX protocol, few adverse effects were noted, and the severity scoring was comparable to that of the vehicle mice (Appendix A). We then determined whether RTX had the expected effect of decreasing the sensory nociception, through evaluating paw withdrawal in the hot plate assay. The RTX-pre-treated mice remained on the hot plate without sensing heat until they were removed after the 20 s cut off period (Appendix A), substantially different from their response before RTX (Appendix A). 

These mice were then examined for their responses to the i.d. injected inflammatory agents. Firstly, to determine if RTX pre-treatment had any functional influence on the cutaneous microvasculature, we investigated the effect of SP + CGRP co-injected i.d, as a positive control due to their oedema-inducing effects previously observed (Figure 1b). SP and CGRP were found to induce similar responses in the RTX pre-treated as vehicle-pre-treated mice (Figure 3b) with no neutrophil accumulation (Figure 3c). We also investigated the response to TNFα and observed the expected significant oedema formation when compared to the control (Tyrode) over 4 h (Figure 3d). However, the response was significantly less in the RTX-pre-treated mice (Figure 3d). By comparison, carrageenan (a general activator of endogenous inflammatory pathways) induced oedema in the vehicle pre-treated mice. RTX pre-treatment led to significantly less carrageenan-induced oedema, and neutrophil accumulation (Figure 3d,e). These findings—that TNF-α and carrageenan-induced oedema were both significantly reduced in the RTX-pre-treated mice—suggest a potential role of the neuropeptides in the inflammatory response. This may include substance P, in addition to CGRP.

### 2.4. The Effect of CGRP Antagonist BIBN4096BS on TNFα and Carrageenan-Induced Oedema Formation and Neutrophil Accumulation

To probe the involvement of CGRP more precisely, we designed experiments using the selective and potent CGRP receptor antagonist (BIBN4096BS) and αCGRP knockout (KO) mice. BIBN4096BS (3 mg/kg i.p. −30 min), a selective antagonist of CGRP responses in mouse skin [9,27], antagonised the hypotensive effect of administration of CGRP (1 nmol/kg i.v.) to an anaesthetised mouse after carotid artery cannulation, as expected (Appendix A). For this study, the CGRP antagonist was required to be active for the entire 4 h experimental period. Thus, an experiment was designed where BIBN4096BS (3 mg/kg i.p.) was administered 30 min before the start of the experiment and then midway through the experimental period (at 2 h). The effectiveness of this administration strategy was investigated by testing the ability of CGRP to potentiate SP-induced oedema formation during the last 0.5 h of the 4 h experimental period (Figure 4a). Results show that in mice treated with vehicle, CGRP was able to potentiate oedema formation as expected. However, in the presence of BIBN4096BS, the response was inhibited back to the level observed with substance P alone (Figure 4b), indicating an effective blockade of CGRP peripheral microvascular responses by using this regime of BIBN4096BS. 

We then proceeded to investigate the effect of TNFα and carrageenan in BIBN4096BS-pretreated mice. Both carrageenan-induced oedema formation and neutrophil accumulation were reduced in mice which had received BIBN4096BS (Figure 5d,e). There was a trend towards reduced TNFα oedema and neutrophil accumulation by BIBN4096BS, but this did not reach significance (Figure 5d,e). 

### 2.5. The Effect of Deleting αCGRP Peptide on TNFα and Carrageenan-Induced Oedema Formation and Neutrophil Accumulation

Finally, experiments were carried out where the responses were investigated in αCGRP KO mice (Figure 6a). The results show that TNFα responses were not affected by αCGRP deletion (Figure 6b,c). Here we carried out a dose–response curve to carrageenan. The response to the highest dose of carrageenan tested (100 µg) was significantly less in αCGRPKO mice, in terms of both oedema formation and neutrophil accumulation (Figure 6d,e). Of note, CGRP mRNA expression was increased in the DRG of WT mice that had received carrageenan (Appendix A).

## 3. Discussion

Initially, we confirmed the oedema-inducing effect of SP and the potentiating effect of vasodilator doses of CGRP, which had no effect on oedema formation when injected alone. Neither peptide had any effect on neutrophil accumulation. The oedema results are in keeping with previous findings [8,9,28], and further supports the potentiating or enhancing role of CGRP at the microvascular level in terms of promoting oedema formation, as a consequence of its vasodilator activity. This is likely to occur at the arterial level, but venules may be involved [2]. This potentiation was also seen when CGRP was injected into sites previously pre-treated (3.5 h before) with TNFα; a period sufficient to allow the upregulation of neutrophil/endothelial adhesion molecules by TNFα, that contribute to neutrophil accumulation [17,18]. Additionally, CGRP did not significantly affect neutrophil accumulation induced by TNFα. We consider that CGRP in vasodilator quantities was able to potentiate oedema formation when injected at 3.5 h, as the neutrophils that accumulate in response to TNFα had an effect in mediating increased microvascular permeability to make the post-capillary venules leaky. The mechanisms behind the TNFα-induced increase in microvascular permeability are likely to be neutrophil-dependent [20] and potentially due to the release of neutrophil-derived agents [29]. TNFα has been suggested to act to acutely increase microvascular permeability over a few minutes [18], but we found no evidence of this. By comparison, endothelial cells in culture respond to TNFα by increasing permeability, independently of neutrophil [30]. This is possibly as the culturing process leads to cells being primed. CGRP can also activate endothelial cells in vitro but this is associated with proliferation and angiogenesis [31]. CGRP in bovine endothelial/smooth muscle culture experiments acts to increase cAMP only and not cGMP [32]. However, more recent studies indicate that CGRP can exhibit endogenous GPCR agonist bias in human cardiovascular cells [33]. These combined results emphasise the importance of an in vivo assay. 

The neuropeptides SP and CGRP are contained within vesicles in sensory neurons [34]. RTX, is a diterpene, an extract of the Euphorbia cactus and an ultra-potent analogue of capsaicin, that activates sensory neurons via nociceptive transient receptor potential vanilloid 1 (TRPV1) receptor [35]. This activation is followed by depletion/desensitisation with an associated loss of sensory nerve function [36]. RTX is used here as a pharmacological tool to deplete the biological active agents contained in sensory neurons. This includes peptides such as somatostatin, in addition to SP and CGRP. It is used here to compare inflammatory responses in the absence and presence of an intact sensory neuronal network (similar to [26]). The neurotoxic effect of RTX can be accompanied by potentially fatal side effects (e.g., lung dysfunction and hypothermia) and pre-treatment with bronchodilators and recovery in a warm environment has been utilised [37,38]. In this study, in order to minimise animal loss and manage animal suffering we co-injected daily anti-inflammatory/analgesic drugs meloxicam, mepyramine and methysergide, each chosen for their distinct profile [39,40,41]. The corticosteroid dexamethasone suppresses inflammation and pain via multiple activities [42] and was injected on the first day only of the RTX protocol. The highest incidence was seen on day one, one hour after RTX administration, where the major adverse effect was loss of mobility, with some breathing dysregulation. The successful depletion of the sensory neuronal pathway was determined using a hot plate test (55 °C) where RTX pre- treated mice were unresponsive. This confirmed the loss of sensory nerve function in RTX-treated mice. 

After confirming sensory nerves depletion, by use of a hot plate, we proceeded with the primary aim of the experiment, to elucidate how sensory nerve depletion influenced the cutaneous vascular responses. An experiment was designed with TNFα and carrageenan over the 4 h period, necessary to assess neutrophil accumulation, in addition to the oedema formation with SP + CGRP as a positive control. The results from this positive control provided evidence that RTX had no damaging/toxic effect on the microvascular lining, as both groups of mice responded similarly to exogenous SP + CGRP. The inflammatory mediators, carrageenan and TNFα, both induced a significant increase in oedema volume in the vehicle mice. Conversely, RTX pre-treated mice showed a significant reduction in the observed oedema volume. Additionally, neutrophil accumulation was significantly reduced at the carrageenan site in the RTX-treated mice whilst the RTX study provided evidence that sensory nerves are involved in the response, this did not allow a clear identification of a role for CGRP.

The effect of the CGRP receptor antagonist (BIBN4096BS) was examined in terms of the responses to TNFα and carrageenan. The significant loss of oedema formation and neutrophil accumulation in response to carrageenan provides direct evidence of a modulatory role of CGRP in this response. This finding was duplicated in αCGRPKO mice that received carrageenan. An overview of these results is shown in Table 1. carrageenan is a seaweed extract that activates a variety of mediator systems. We have previously shown an involvement of substance P in carrageenan-induced inflammation in mice [22]. The role of CGRP in carrageenan-induced inflammation has not been fully investigated, although the hyperalgesic role of CGRP in the carrageenan model is clear [43]. There is evidence from the rat that suggests that CGRP is not involved in carrageenan-induced inflammation [44]. However, a role in carrageenan-induced oedema formation and neutrophil accumulation has not been previously studied in mouse skin to our knowledge. It is known though that carrageenan paw tissue oedema is reduced in the same colony of αCGRPKO mice as used in this study, but the mechanisms were not investigated [45]. It was suggested that this was linked to the ability of αCGRP to be released peripherally to influence the inflammatory process, in terms of tissue swelling. Perhaps surprisingly, picomolar amounts of CGRP has been found to induce oedema when injected alone or to contribute to oedema when injected with carrageenan in the oro-facial tissue of the rat [46]. This was suggested to be partially due to 5-HT released from mast cells, but other mechanisms remain unclear [46].

A pro-inflammatory role of CGRP in neutrophil accumulation, one of the first cells to reach an inflammatory site [47], has been suggested by in vitro studies [48,49]. However, the most physiologically relevant evidence of a potentiation of vascular inflammation comes from in vivo situations where CGRP is acting as a vasodilator to potentiate neutrophil-dependent oedema formation induced by IL-1β (e.g., [8,50,51]) and now, as shown here for TNFα. We did not gain evidence for a direct influence of CGRP on TNFα-induced neutrophil accumulation, although blockade of CGRP was linked to a loss of neutrophil accumulation induced by carrageenan. There are suggestions that CGRP enhances and inhibits neutrophil accumulation depending on the situation. For example, CGRP has been suggested to upregulate Intracellular Adhesion Molecule-1 (ICAM-1) expression and neutrophil accumulation in the eye, infected with Pseudomonas aeruginosa infection, which consequently improved the prognosis of the disease [52]. Additionally, mice with the receptor activity modifying protein 1, RAMP1, deleted were used to examine the role of CGRP in acute pancreatitis. The RAMP1KO mice showed an increased infiltration of neutrophils and oedema formation. This led to an accelerated progression of pancreatitis symptoms, suggesting that CGRP was acting in a protective manner, which may well be linked to neutrophil accumulation [53]. Considering the present experiments in the cutaneous microvasculature, certainly the results suggest the ability of CGRP to potentiate acute inflammation. Whilst the vasodilator dose of CGRP used, leads to our understanding that the potentiation is a consequence of vasodilator activity; it is also important to consider further possibilities. It has been suggested that CGRP, through a mechanism involving macrophages, can reduce clearance of exudate by lymph glands, which would lead to increased oedema volume [54]. There are observations of protective roles of CGRP, that may be due to its vasodilator activity, in wound healing [55] and scleroderma [55], but of pro-inflammatory pathways, in terms of a role in T cell switching involving the IL-17 pathway to regulate cutaneous immunity [56] of possible relevance to psoriasis, where CGRP levels have been reported to be elevated [57]. Recent work has also shown that, by obstructing CGRP- mediated IL-23 production from psoriatic dendritic cells, lidocaine has the ability to ameliorate IMQ-induced psoriasis-like skin inflammation [58].

To conclude, evidence is provided that CGRP can potentiate the ability of TNFα to induce inflammatory swelling in the skin. Additionally, endogenous CGRP contributes to oedema formation and neutrophil accumulation induced in the carrageenan model of inflammation. Thus, we suggest that blockade of CGRP, such as achieved through use of a CGRP antagonist, has the potential to lessen inflammatory conditions in skin.

## 4. Materials and Methods

### 4.1. Mice

A total of 43 CD1 mice (male, 30–35 g, 8–12 weeks; Charles River, UK) and 40 wildtype (WT) and αCGRP knockout (αCGRPKO) C57BL/6 mice (bred in house) were used. They were kept on a normal diet and allowed free access to food and water, and mice were housed under standard conditions with a 12-h light/dark cycle. The WT and αCGRPKO mice were age- and sex-matched with littermates. The αCGRPKO mice were originally a gift from Prof. A. M. Salmon and generated as previously described here [59]. Briefly, exon 5, which represents the αCGRP coding region within the calcitonin gene, was replaced by a cassette containing Lac Z/CMV/neomycin resistance genes. Through electroporation, HM-1 embryonic cells were transfected with the construct and then injected into blastocytes of C57BL/6 mice. Heterozygous αCGRP mice were obtained following germline passage and crossing with the C57BL/6 strain, and by pairing heterozygous mice, homozygous αCGRP KO and WT mice were generated. All experimental groups were randomly allocated. Experiments were performed according to the Animals (Scientific Procedures) Act 1986 and the King’s College Animal Care and Ethics Committee in keeping with the ARRIVE guidelines.

### 4.2. In Vivo Dorsal Skin Oedema Inflammation Model 

Mice were weighed and anaesthetised with 5% isoflurane (Isocare; Animalcare, York, UK) carried in 1 L/min oxygen (maintained with 2% isoflurane). The dorsal skin was shaved, and a balanced site pattern of injection sites marked out, method adapted from Sawyer et al., (2011) originally described as the Miles assay [8,60] and further developed and validated by Zarban et al., 2022 [61]. Sites were marked out according to a balanced site design for intradermal (i.d.) injections (50 μL) using a 27G needle. Prior to intradermal injections, Evans Blue, (1.25% in saline, 0.1 mL/mouse), Sigma-Aldrich, Dorset, UK) was injected i.v. Evans Blue binds to serum albumin and was used as a marker for areas of oedema formation. After randomly assigning agents, the skin was injected with SP (Sigma-Aldrich) (300 pmol), CGRP (AnaSpec Inc., Fremont, CA, USA) (20 pmol), TNF-α (NovusBio, Centennial, CO, USA) (100 ng), carrageenan Lambda (Sigma-Aldrich., Dorset, UK) (10, 30 and 100 μg) and vehicles, depending on protocol. Agents were dissolved in Tyrode’s solution (Tyrode’s solution: 137 mM NaCl, 2.68 mM KCl, 0.4 mM NaH_2_PO_4_, 11.9 mM NaHCO_3_, 0.5 mM MgCl_2_, and 5.6 mM glucose in distilled water). Tyrode’s alone was used as the vehicle control for i.d. injections. Un-injected and sham-injected sites (that received injection needle prick, but no fluid) were also investigated. Mice were allowed to recover when necessary and, at the pre-designated time, re-anaesthetised and humanely killed by cervical dislocation. The dorsal skin was carefully blunt dissected, collected and pinned non-fur side up, without excessive pulling to allow clearer visualisation of the blue oedema ‘bleb’. Two perpendicular diameters (length and width) of the oedema formed were measured in mm using a ruler and the oedema depth was measured in mm by placing the oedema site within an engineer’s calliper (Moore and Wright, Camberley, UK). The calliper was adjusted, until it touched, but did not squeeze both surfaces of the skin area. The calliper was then locked, ensuring the measurement was stable when taking the skin away. The three diameters (length, width and depth) of the ellipsoid shaped oedema were then determined, using the following equation to calculate oedema volume: oedema volume (mm³) = (π/6) × length × width × depth.

### 4.3. Obtaining an Internal Standard via Zymosan-Induced Peritoneal Inflammation 

In order to develop a reference sample of MPO, a sample was prepared from mouse peritoneal inflammatory exudate. The exudate was obtained through inducing peritoneal inflammation by zymosan (1 mg/0.5 mL i.p.) for 4 h (Adapted from [22]).

### 4.4. Sensory Nerve Depletion—Resiniferatoxin (RTX) Pre-Treatment 

The mice were randomly allocated to vehicle (10% Tween-80, 10% ethanol, 80% saline) or RTX (0.3 mg/kg s.c., Sigma-Aldrich, Dorset, UK) groups. All mice received a single intraperitoneal (i.p.) injection of a mixture of anti-inflammatory agents 30 min before the RTX protocol (all from Sigma-Aldrich, Dorset, UK unless stated). These agents were: the antihistamine mepyramine (20 mg/kg), the cyclo-oxygenase non-steroidal anti-inflammatory drug meloxicam (2 mg/kg, Boehringer Ingelheim, Ingelheim, Germany) and the anti-5-HT antagonist methysergide (2 mg/kg). They were administered 30 min before the start of the protocol. The anti-inflammatory steroid dexamethasone (5 mg/kg i.p., Duphacort, Zoetis, London, UK) was administered on day one, 30 min–1 h after RTX injection. A distress scoring system was used to record the physical response of the mice following RTX/vehicle pre-treatment, with scores 0 (normal)−4 (severe adverse effect) for each of loss of immobility and observation of behaviour/stress changes. Mice were observed immediately after RTX or vehicle injections for 4 h. Food and water intake were also monitored and scored between zero to three with zero being a normal intake (no change) and three being a severe response to RTX/vehicle. 

### 4.5. Evaluation of Sensory Nerve Depletion—Hot Plate Test 

A hot plate test, where mice were placed on a hot plate (55 °C) for a maximum of 20 s, was used to confirm successful sensory nerve denervation. The time taken for mice to show discomfort (withdrawal latency), such as hind paw licking or flinching, was recorded. Mice that did not show discomfort within the timeframe (20 s) were removed from the hot plate to prevent tissue damage [62,63].

### 4.6. Systemic Treatments for Skin Assay 

The CGRP receptor antagonist BIBN4096BS (3 mg/kg i.p., given at −0.5 h and at 2 h in a 4 h protocol; Tocris, Abingdon, UK) or vehicle (3.4% DMSO in saline) was investigated in one set of experiments. 

### 4.7. Ex Vivo Assay of Myeloperoxidase (MPO) Accumulation 

Skin sites were collected from the dorsal skin (10 mm diameter), weighed and snap-frozen in liquid nitrogen, then stored at −80 °C. Skin samples were homogenised in 500 µL/sample (0.1 M NaCl, 0.02 M NaPO_4_, 0.015 M EDTA, 0.5% HTAB, pH 4.7) and lysed using QIAGEN TissueLyser II at 30 Hz for 5 min, three times, cooling between each. Skin samples were then centrifuged at 17,000× *g* for 15 min at 4 °C and the supernatant was collected. Ninety-six-well plates were used for the assay. Neutrophil accumulation was measured by comparing MPO activity in extracts with standards. H_2_O_2_ oxidation of 3,3′, 5,5′-Tetramethylbenzidine (TMB) Liquid Substrate System, (Sigma-Aldrich, Dorset, UK) was used to determine MPO activity. In a 96-well plate, 25 μL of MPO buffer was added to 25 μL of sample and 100 μL of TMB liquid substrate then added. The plate was incubated in the dark at 37 °C for 15 min. Absorbance (OD) was read at 620 nm using a spectrophotometer and a standard curve was plotted of OD against MPO in the standard samples and calculated as U/mL.

### 4.8. Quantitative Polymerase Chain Reaction 

Mouse DRGs were prepared and analysed as in reference [64]. Total RNA was isolated from DRGs using the RNeasy Micro Kit (#74004, Qiagen, Crawley, UK) and 500 ng of purified RNA was reverse transcribed into cDNA using SuperScript ViLO cDNA synthesis kit (#11754050, Thermo Fisher Scientific, Dartford, UK). qPCR was performed with 10 ng of cDNA using the PowerUp SYBR Green master mix kit (#A25780, Thermo Fisher Scientific, Dartford, UK) in the 7900HT Real-Time PCR machine (Applied Biosystems, Waltham, MA, USA). Gene expression was quantified, and the double delta CT analysis method was used for relative fold-change of mRNA expression levels. CGRP primers used: forward primer: CTGAGGGCTCTAGCTTGGAC, reverse primer: TGCCAAAATGGGATTGGTGG (Sigma).

### 4.9. Measurement of Blood Pressure via Carotid Artery Cannulation 

Systemic pressure was monitored in a similar manner to previously [6]. The mouse was anaesthetised using isoflurane (2%, Abbott laboratories, Maidenhead, UK) in 0.5 L/min O_2_ and core body temperature was maintained at 37 °C with a homeothermic blanket (Harvard Apparatus, Cambridge, UK). After isolating the left carotid artery, a cannula (Smiths Medical-Portex, Hythe, UK) filled with heparin—100 U/mL diluted in 0.9% saline—was introduced into the carotid artery. The mouse was left to stabilise for 5 min before treatments. Systemic pressures were measured using the PowerLab data acquisition system and LabChart 8 Pro software (ADInstruments Ltd., Oxford, UK).

## 5. Data Analysis

Results are shown as mean ± SEM and were analysed by one-way or two-way ANOVA followed by Bonferroni’s post hoc test. Statistics were performed using GraphPad Prism software (version 8.4.3). A probability level of 0.05 or smaller was used to indicate statistical significance.

## Figures and Tables

**Figure 1 ijms-23-12246-f001:**
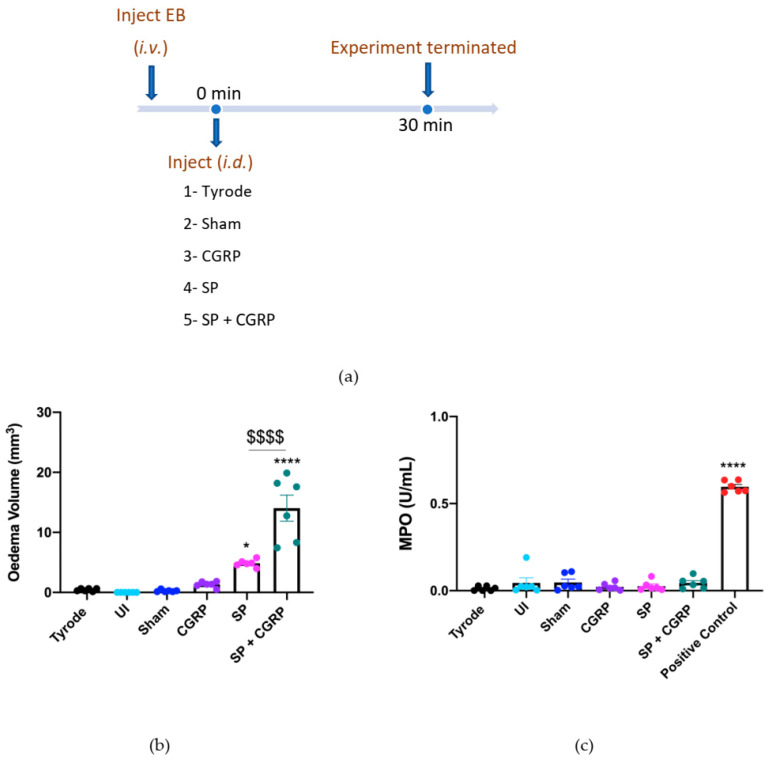
Calcitonin gene-related peptide, CGRP, potentiates oedema formation induced by substance P, SP, over 30 min. (**a**) Timeline for studying the effect of exogenous CGRP on SP-induced oedema over 30 min. Oedema volume (**b**) and neutrophil accumulation (**c**) measured 30 min after administering into dorsal skin sites either controls, Tyrode (50 µL) or sham (0 µL); or test agents, CGRP (20 pmol/50 µL), SP (300 pmol/50 µL) or SP (300 pmol/50 µL) with CGRP (20 pmol/50 µL i.d.). An internal standard (zymosan-induced peritoneal exudate) was used as a positive control for MPO. Results are shown as mean ± SEM (*n* = 6). One-way ANOVA with Bonferroni’s post hoc test. * *p* < 0.05, **** *p* < 0.0001 are compared to Tyrode. ^$$$$^ *p*< 0.0001 is compared to SP.

**Figure 2 ijms-23-12246-f002:**
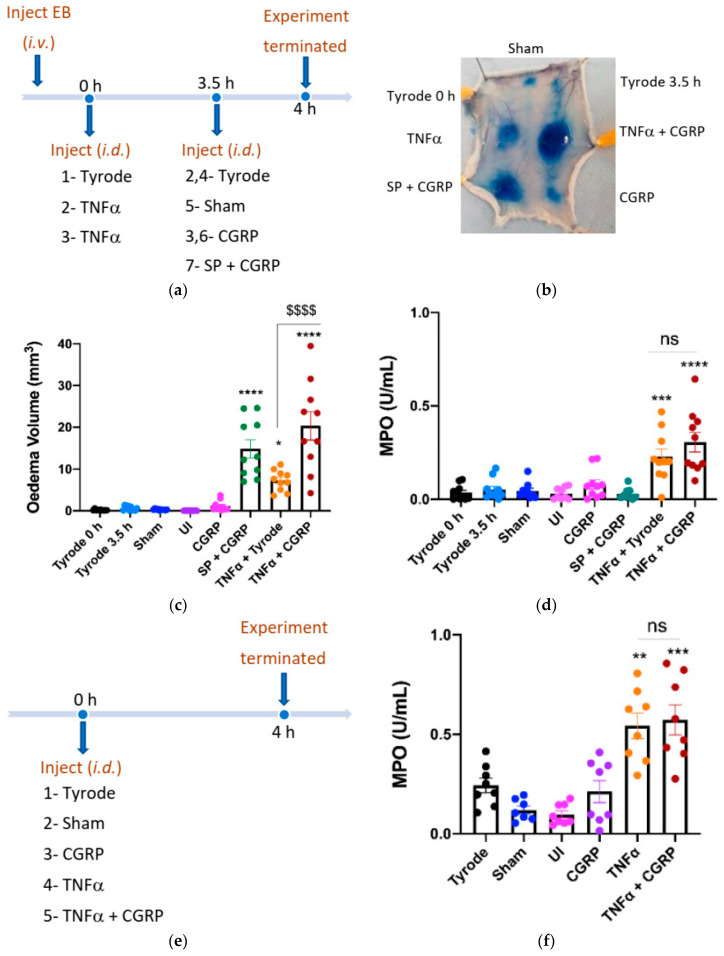
Calcitonin gene-related peptide, CGRP (when administered i.d. for the last 30 min) potentiates oedema formation, but not neutrophil accumulation induced by tumour necrosis factor alpha, TNFα, over 4 h. (**a**) Timeline for studying the effect of CGRP on TNFα-induced inflammation over 4 h, with CGRP or control added i.d. at the last 30 min (as c, d). (**b**) Representative image. Oedema volume (**c**) and neutrophil accumulation (**d**) were measured. This followed injection of TNFα (100 ng/50 µL i.d.) or vehicle (Tyrode) at 0 h; then after 3.5 h vehicle (Tyrode) or CGRP (20 pmol/50 µL i.d.) were injected at the TNFα sites. SP (300 pmol/50 µL) was also injected with CGRP (20 pmol/50 µL i.d.) at 3.5 h as a positive control for acute oedema formation. Results are shown as mean ± SEM (*n* = 10). One-way ANOVA with Bonferroni’s post hoc test. * *p* < 0.05, *** *p* < 0.001, **** *p* < 0.0001 are compared to Tyrode 0 h or Tyrode 3.5 h. ^$$$$^ *p* < 0.0001 is compared to TNFα + Tyrode. (**d**) Timeline for studying effect of CGRP when injected with TNFα at 0 h, for 4 h. (**e**) Timeline for studying the effect of CGRP on TNFα-induced inflammation over 4 h, with CGRP co-injected with TNFα i.d. at 0 h. (**f**) Neutrophil accumulation measured after co-injecting CGRP (20 pmol/50 µL i.d.) with TNFα (100 ng/50 µL i.d.) for 4 h. Results are shown as mean ± SEM (*n* = 8). One-way ANOVA with Bonferroni’s post hoc test. ** *p* < 0.01, *** *p* < 0.001 are compared to Tyrode.

**Figure 3 ijms-23-12246-f003:**
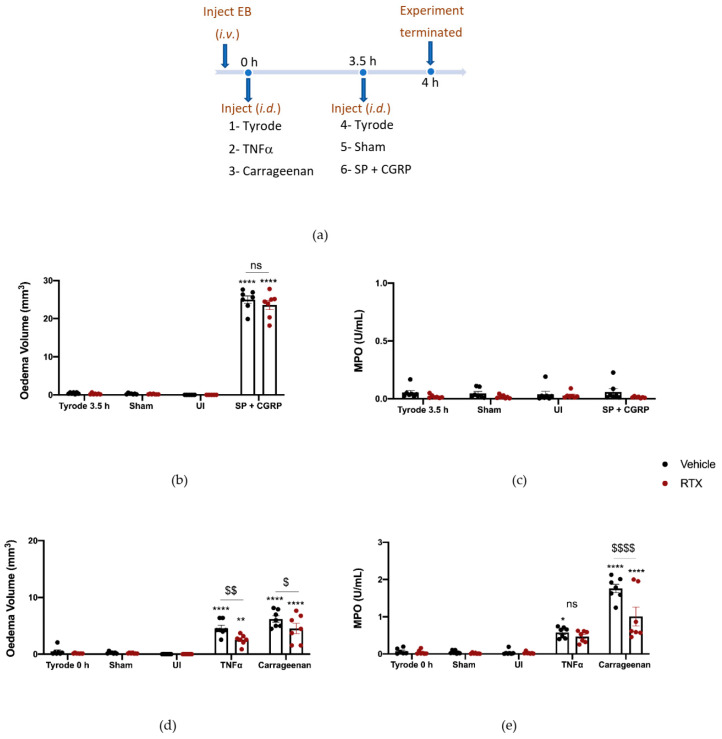
Sensory nerve depletion reduces oedema formation induced by tumour necrosis factor alpha, TNFα, and carrageenan, but only neutrophil accumulation induced by carrageenan over 4 h. (**a**) Timeline for studying the effect of sensory nerve depletion on TNFα and carrageenan-induced inflammation over 4 h in Resiniferatoxin (RTX)-pre-treated mice and vehicle-pre-treated mice. Oedema volume (**b**) and neutrophil accumulation (**c**) measured after injecting SP (300 pmol/50 µL) with CGRP (20 pmol/50 µL i.d.) for the last 30 min of the protocol. (**d**) Oedema volume and (**e**) neutrophil accumulation measured 4 h after injecting TNFα (100 ng/50µL i.d.) and carrageenan (100 µg/50 µL i.d.) in vehicle or RTX (0.3 mg/kg s.c.) pre-treated mice. Results are shown as mean ± SEM (*n* = 7). Two-way ANOVA with Bonferroni’s post hoc test. * *p* < 0.05, ** *p* < 0.01 and **** *p*< 0.0001 are compared to Tyrode 0 h or Tyrode 3.5 h. ^$^ *p* < 0.05, ^$$^ *p* < 0.01 and ^$$$$^ *p* < 0.0001 are compared to Vehicle TNFα or carrageenan.

**Figure 4 ijms-23-12246-f004:**
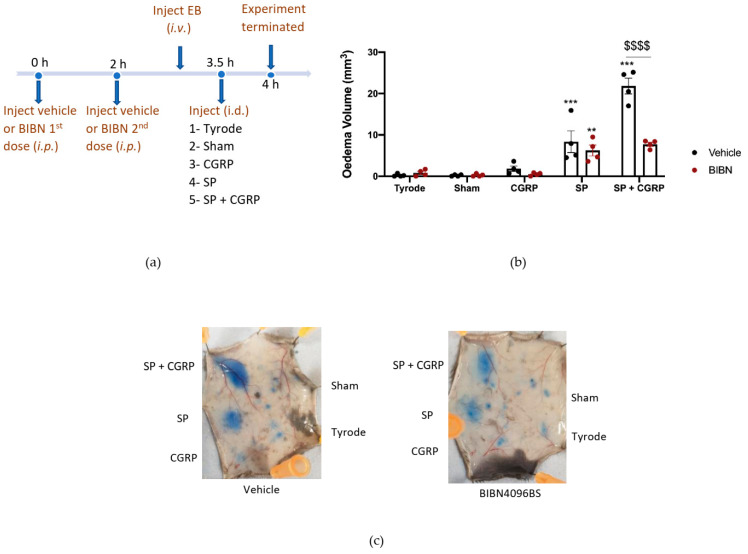
Characterisation of the Calcitonin gene-related peptide, CGRP, receptor antagonist, BIBN4096BS, and effect on oedema formation induced by SP + CGRP for 30 min after 3.5 h pre-treatment. (**a**) Timeline for studying the effect of the CGRP receptor antagonist BIBN4096BS (3 mg/kg i.p., given at −0.5 h and at 2 h) over 4 h, on oedema formation induced by substance P (SP), CGRP, as well as SP + CGRP over 30 min. (**b**) Oedema volume measured after injecting SP (300 pmol/50 µL) ± CGRP (20 pmol/50 µL i.d.) for the last 30 min of the protocol in vehicle or BIBN4096BS pre-treated mice. (**c**) Representative images. Results are shown as mean ± SEM (*n* = 4). Two-way ANOVA with Bonferroni’s post hoc test. ** *p* < 0.01 and *** *p* < 0.001 are compared to Tyrode. ^$$$$^ *p* <0.0001 is compared to Vehicle SP + CGRP.

**Figure 5 ijms-23-12246-f005:**
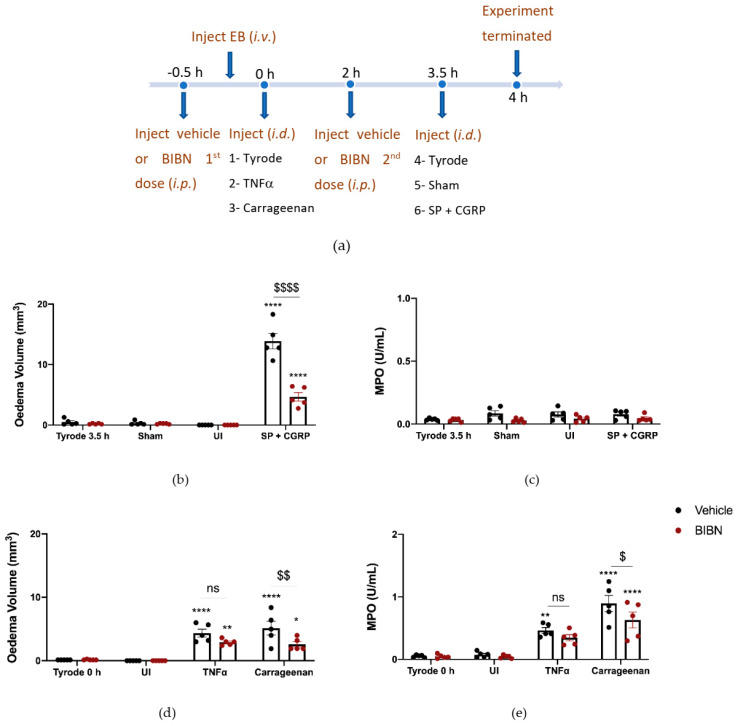
Calcitonin gene-related peptide, CGRP, receptor antagonist, BIBN4096BS, reduces oedema formation and neutrophil accumulation induced by carrageenan but not tumour necrosis factor alpha (TNFα)-induced inflammation over 4 h. (**a**) Timeline for studying the effect of the CGRP antagonist BIBN4096BS (3 mg/kg i.p., given at −0.5 h and at 2 h) on TNFα and carrageenan-induced inflammation over 4 h. Oedema volume (**b**) and neutrophil accumulation (**c**) measured after injecting the positive control SP (300 pmol/50 µL) with CGRP (20 pmol/50 µL i.d.) at the last 30 min of the protocol. (**d**) Oedema volume and (**e**) neutrophil accumulation measured 4 h after injecting TNFα (100 ng/50 µL i.d.) and carrageenan (100 µg/50 µL i.d.) in vehicle or BIBN4096BS pre-treated mice. Results are shown as mean ± SEM (*n* = 5). Two-way ANOVA with Bonferroni’s post hoc test. * *p* < 0.05 and ** *p* < 0.01 and **** *p* < 0.0001 are compared to Tyrode 0 h or Tyrode 3.5 h. ^$^ *p* <0.05, ^$$^ *p* <0.01 and ^$$$$^ *p* <0.0001 are compared to Vehicle SP + CGRP or Carrageenan.

**Figure 6 ijms-23-12246-f006:**
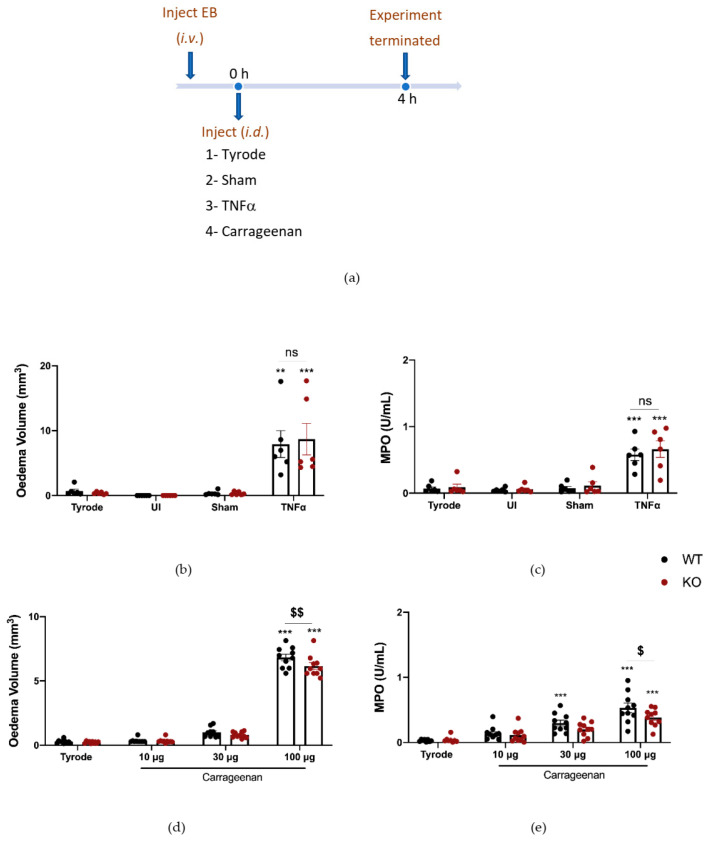
Oedema formation and neutrophil accumulation induced by carrageenan over 4 h is reduced in αCGRPKO mice. (**a**) Timeline for studying tumour necrosis factor alpha (TNFα) and carrageenan-induced inflammation over 4 h in WT and αCGRPKO mice. Oedema volume (**b**) and neutrophil accumulation (**c**) measured 4 h after injecting TNFα (100 ng/50 µL i.d.). Results are shown as mean ± SEM (*n* = 6). (**d**) Oedema volume and (**e**) neutrophil accumulation measured 4 h after injecting carrageenan (10, 30 and 100 µg/50 µL i.d.) in WT or αCGRPKO mice. Results are shown as mean ± SEM (*n* = 10). Two-way ANOVA with Bonferroni’s post hoc test. ** *p* < 0.01 and *** *p* < 0.001 are compared to Tyrode. ^$^ *p* < 0.05, ^$$^ *p* < 0.01 are compared to Vehicle carrageenan 100 μg.

**Table 1 ijms-23-12246-t001:** Summary data to show the effect of modulating Calcitonin gene-related peptide, CGRP, levels on oedema and neutrophil accumulation (Myeloperoxidase, MPO) in response to carrageenan and tumour necrosis factor alpha, TNFα. Three models that enable modulation of CGRP activity were used. Sensory nerves were depleted with resiniferatoxin (RTX 0.3 mg/kg, s.c. for 4 days), left hand columns. CGRP responses were antagonised with CGRP receptor antagonist, BIBN4096BS (3 mg/kg i.p. given at two timepoints, see methods), middle columns. WT and αCGRP KO mice were also investigated, right-hand columns. For all experiments carrageenan, TNFα and Tyrode (control) were investigated over 4 h. Results are shown as mean ± SEM (*n* = 5–10). Two-way ANOVA with Bonferroni’s post hoc test. Significance is shown for agent compared with relevant control; ns is non-significant.

Expt Aim	Deplete Neuropeptides	Deplete Neuropeptides	Antagonise CGRP	Antagonise CGRP	CGRP Lacking	CGRP Lacking
Treatments/ Assays	RTX Oedema	RTX MPO	BIBN4096 Oedema	BIBN4096 MPO	CGRPKO Oedema	CGRPKO MPO
Tyrode	ns	ns	ns	ns	ns	ns
Carrageenan	*p* < 0.05	*p* < 0.001	*p* < 0.01	*p* < 0.05	*p* < 0.01	*p* < 0.05
TNFα	*p* < 0.01	ns	ns	ns	ns	ns

## Data Availability

Not applicable.

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
