# Peer review of "Elucidating the Ability of CGRP to Modulate Microvascular Events in Mouse Skin"

_ijms, 2022, doi:10.3390/ijms232012246_

Round 1

Reviewer 1 Report

The article by Zarban et al focuses on the ability of CGRP to modulate microvascular events in mouse skin, with the use of intradermal injections. Although the paper is interesting, I have some methodological questions. Further, although the English quality is high, the structure of writing does not follow the same quality.

The Paper needs significant rewriting, there is very little difference between the results section and the first 3 pages of discussion. Either the results need to be more to the point referring to the data etc. and maintain the discussion. Or the discussion should be shorted at least 50%, now it feels like you read the same text twice…

The legends of the figures are not well written, for example, legend figure 3: “when given for the last 30 min” when was it given as an infusion over 30 minutes? or was it given 30 minutes before the sacrifice of the mouse? Further “measured 4h after injecting TNFa (…) for 3.5 h before adding Tyrode or CGRP (…) for the last 30 min in vehicle or meloxicam (…) pre-treated mice”, it is not possible to understand this wording, you inject for 3.5 hours?? Then when you read it first appears that CGRP is in vehicle, but then this refers to the mouse after the brackets. This is just an example, but all legends have similar problems…

The use of the wording “animal culled” is new to me, when I look it up it is usually used as “artificial equalization of the number of offspring in a litter (culling)” but this is not what you do, “sacrifice” might me better?

It is not clear was mouse strain you term CGRO KO, you write “bred in house” what is the strain name, was is purchased, gifted or generated in house? If generated, how was it generated? Further, for clarity do not “denote the αCGRP KO as CGRP KO, this is not correct, use αCGRP KO

The ARRIVE guidelines are reporting guidelines, not guidelines for experiments. And you do not keep the guidelines, as an example you do not report housing of animals, neither do you report blinding, and the list continues, I suggest you read the actual guidelines for reporting…  https://arriveguidelines.org/ Guidelines also suggest reporting background for group size, what is your argument for the highly variable group sizes? Further in line with the 3Rs and reduction in animal use, the reviewer questions if it was necessary to use RTX (very stressful), CGRP antagonism (BIBN) and αCGRP KO animals… Please justify what these groups add in value. Please report the total number of animals used in this study.

The quantification of oedema would be useful to include a sample picture. When you inject EB, the whole mouse turns blue I presume, how do you determine oedema? In all your data you have several groups, are these individual animals or do you inject separate places on the same animal? Does this mean that animals in figure 2 get more injections that in figure 1? When you combine substances do you keep or change the volume (e.g. line 102, did they get 100ul, if yes, did you inject vehicle in the CGRP and SP groups, or did they only get 50 ul? Some concentrations are written as mg/ml such as zymosan, then you also need to state volume.

Line  456, “5% isoflurane in 1% oxygen”? Cannot be correct??

Reviewer 2 Report

The authors provided evidence that applied CGRP, being ineffective alone, can potentiate oedema formation induced by SP or TNF-alfa. In addition, endogenous CGRP contributes to inflammatory oedema induced by TNF-alfa or carrageenan.

This is a well written manuscript with multiple experimental paradigms that have provided coherent results.

 Abstract

The first word could be deleted.

TNF-alfa should be written out (not only the abbreviation).

 Introduction

Line 43: „sensory nerves in peripheral vasculature” is confusing.

Line 54: the word „inflammatory” could be deleted.

 Results:

In all cases when a synergistic response is mentioned (e.g. in line 104), it should be emphasized that the combined effect was higher than the sum of the two individual ones, therefore potentiation or synergism has been revealed.

Line 121: instead of „role”, „effect” should be written.

Line 148: instead of „in”, „to” should be written.

The results regarding the role of prostaglandins do not fit to the main aim of the study.

In several figures the Y axis is unnecessarily too long (e.g. Figs. 2 and 3).

 Discussion

The very long discussion should be shortened considerably (at least by one third).

Although it is clear that CGRP can potentiate oedema formation induced by SP or TNF-alfa, I do not see evidence for the involvement of precapillary arterial vasodilatation in this response. CGRP might exert its effect on the postcapillary venules as well. This problem should be discussed.

Regarding results obtained with RTX treatment, it should be mentioned in the discussion that not only SP and CGRP are stored in peptidergic nerve endings, e.g. somatostatin having anti-inflammatory effects is also present and depleted by RTX.

Line 328: TRPV1 should be written out (not only the abbreviation).
